# Mismatch between Tissue Partial Oxygen Pressure and Near-Infrared Spectroscopy Neuromonitoring of Tissue Respiration in Acute Brain Trauma: The Rationale for Implementing a Multimodal Monitoring Strategy

**DOI:** 10.3390/ijms22031122

**Published:** 2021-01-23

**Authors:** Mario Forcione, Mario Ganau, Lara Prisco, Antonio Maria Chiarelli, Andrea Bellelli, Antonio Belli, David James Davies

**Affiliations:** 1Neuroscience and Ophthalmology Research Group, Institute of Inflammation and Ageing, College of Medical and Dental Sciences, University of Birmingham, Edgbaston, Birmingham B15 2TT, UK; a.belli@bham.ac.uk (A.B.); daviesdj@doctors.org.uk (D.J.D.); 2Department of Neurosurgery, University Hospitals Birmingham NHS Foundation Trust, Mindelsohn Way, Birmingham B15 2TH, UK; 3Department of Clinical Neurosciences, Oxford University Hospitals NHS Foundation Trust, Headly Way, Oxford OX3 9DU, UK; mario.ganau@alumni.harvard.edu (M.G.); lara.prisco@ndcn.ox.ac.uk (L.P.); 4Imaging and Clinical Sciences, Department of Neuroscience, University G. D’Annunzio of Chieti-Pescara, Institute for Advanced Biomedical Technologies, Via Luigi Polacchi 13, 66100 Chieti, Italy; antonio.chiarelli@unich.it; 5Department of Biochemical Sciences “A. Rossi Fanelli”, Faculty of Pharmacy and Medicine, Sapienza University of Rome, Viale Regina Elena 332, 00185 Rome, Italy; andrea.bellelli@uniroma1.it; 6National Institute for Health Research Surgical Reconstruction and Microbiology Research Centre (NIHR- SRMRC), University Hospitals Birmingham NHS Foundation Trust, Mindelsohn Way, Birmingham B15 2TH, UK

**Keywords:** tissue partial oxygen pressure, near-infrared spectroscopy, multimodal neuromonitoring, biosignature, traumatic brain injury, tissue respiration, contrast-enhanced near-infrared spectroscopy, diffuse optical tomography, computerized tomography, microdialysis

## Abstract

The brain tissue partial oxygen pressure (PbtO_2_) and near-infrared spectroscopy (NIRS) neuromonitoring are frequently compared in the management of acute moderate and severe traumatic brain injury patients; however, the relationship between their respective output parameters flows from the complex pathogenesis of tissue respiration after brain trauma. NIRS neuromonitoring overcomes certain limitations related to the heterogeneity of the pathology across the brain that cannot be adequately addressed by local-sample invasive neuromonitoring (e.g., PbtO_2_ neuromonitoring, microdialysis), and it allows clinicians to assess parameters that cannot otherwise be scanned. The anatomical co-registration of an NIRS signal with axial imaging (e.g., computerized tomography scan) enhances the optical signal, which can be changed by the anatomy of the lesions and the significance of the radiological assessment. These arguments led us to conclude that rather than aiming to substitute PbtO_2_ with tissue saturation, multiple types of NIRS should be included via multimodal systemic- and neuro-monitoring, whose values then are incorporated into biosignatures linked to patient status and prognosis. Discussion on the abnormalities in tissue respiration due to brain trauma and how they affect the PbtO_2_ and NIRS neuromonitoring is given.

## 1. Introduction

Currently, there is no disease-modifying treatment for the primary brain injuries sustained following a traumatic event. Instead, the treatment for moderate/severe traumatic brain injury (TBI) is limited to the prevention of a subsequent pathological insult (e.g., intracranial hypertension, hypoxemia, hypotension, cerebral herniation), which is known as a secondary brain injury [1]. Pivotal components of established treatment strategies in TBI care are the continuous neuromonitoring of structural (e.g., hemorrhages, brain edema) and physiological (e.g., intracranial pressure (ICP), tissue partial oxygen pressure (PbtO_2_), lactate/pyruvate ratio) parameters by invasive (e.g., intracranial monitor) and non-invasive (e.g., computerized tomography (CT)) neuromonitoring techniques, and adaptation of the therapy accordingly to maintain intracranial homeostasis.

Near-infrared spectroscopy (NIRS) could provide non-invasive, continuous neuromonitoring of the brain tissue respiration in TBI [2,3]. However, to date, the commercially available and clinically usable NIRS devices (e.g., INVOS 5100 Cerebral Oximeter, ISS-OxiplexTS^TM^) have not shown sufficient ability to detect episodes of ischemia in cases of severe TBI compared to the invasive, intracranial monitor of PbtO_2_ [4,5,6,7]. Due to this low sensitivity to episodes of ischemia, researchers deemed the NIRS technique unsuitable for neuromonitoring as an independent treatment modality and thus an inadequate replacement for the invasive PbtO_2_ neuromonitoring, limiting the use of NIRS to research only [8,9].

In this scoping review, we argue that the values reported by the intracranial monitoring of PbtO_2_ are not completely representative of the brain tissue respiration. The capacity of NIRS to assess parameters indicative of TBI burden, which are not investigated at all by the intracranial PbtO_2_ monitor, cannot be determined by limiting to a crude comparison between the two techniques. Potential advantages of NIRS in brain trauma care may remain unknown and untested to date. Furthermore, the PbtO_2_ and the NIRS values pertain to different subprocesses of tissue oxygenation. These processes are altered in brain trauma: effects differ between patients (interpatient variability); effects also vary for the same patient (intrapatient variability) across different areas of the brain and at different times since injury. As such, the values measured by the intracranial PbtO_2_ monitor and by NIRS diverge, to a greater or lesser extent, according to the type and severity of TBI, as well as to the time of assessment.

The complexity of pathologies in brain trauma indicates that a crude substitution of PbtO_2_ values for those of NIRS is not a valid approach to incorporating the optical techniques into clinical practice. Multiple NIRS techniques (e.g., diffuse optical tomography, contrast-enhanced NIRS) should be included into a multimodal neuromonitoring approach with systemic (e.g., arterial blood gas analysis (ABG)), invasive (e.g., PbtO_2_, microdialysis) and radiological (e.g., CT scan) neuromonitoring devices [10]. Optical biomarkers can be integrated into biosignatures to reflect specific subtypes of pathology.

Physiological tissue respiration and its possible impairment in brain trauma are discussed with the view to analyze the significance of values recorded by the intracranial PbtO_2_ monitor and NIRS to the pathogenesis of acute brain trauma. A detailed model of the tissue respiration and its properties post-TBI would require a more in-depth analysis of all the possible biochemical mechanisms, which is beyond the scope of this review.

## 2. Tissue Respiration

The passage of oxygen (O_2_) from vessels into the cells comprises multiple steps in different volumes (e.g., intracellular erythrocytes, plasma, interstitial tissue, intracellular brain cells) and is influenced by multiple physiological factors (e.g., cerebral blood flow, capillary density, concentration of hemoglobin (Hb), O_2_ affinity for Hb) [11]. The pathogenesis of brain trauma may alter the mechanisms that regulate these steps.

### 2.1. Physiological Tissue Respiration

#### 2.1.1. Oxygen Forms in the Blood and Oxygen Diffusion

In blood vessels, O_2_ is dissolved in its gaseous form and is in equilibrium with the O_2_ bound to the Hb, which is the main component [11,12]. Concentrations of the gas component of O_2_ alone are usually given as arterial partial oxygen pressure (PaO_2_). The O_2_ diffuses across the blood–brain barrier (BBB) in its gaseous form driven by the differences in partial oxygen pressure between the plasma and interstitial tissue [13].

#### 2.1.2. Hemoglobin and Oxygen–Hemoglobin Dissociation Curve

In adults, the majority of Hb is a tetramer compounded by two alpha and two beta subunits, each holding a heme group, which is an organic compound (iron–protoporphyrin IX) with an atom of iron in a ferrous oxidative state (Fe^2+^), where the O_2_ can be bound [11,14]. O_2_ is a homeotropic allosteric modulator to Hb [11,14]. O_2_ binding presents positive cooperativity, and the oxygen–hemoglobin dissociation curve has a sigmoid shape as a result [11,14].

Hydrogen (H^+^), carbon dioxide (CO_2_), and 2,3-diphosphoglycerate (2,3-DPG) are heterotropic allosteric modulators to Hb, and they can shift the sigmoid-shaped oxygen–hemoglobin dissociation curve to the right, i.e., to higher values of O_2_ pressure [15]. Similar results can also be obtained in vitro by increasing temperature.

The Bohr effect is responsible for the shift of the oxygen–hemoglobin dissociation curve caused by changes in H^+^ concentration [15]. CO_2_ has a two-fold role in the shift of the oxygen–hemoglobin dissociation curve: it directly binds to the N-terminal amino groups of Hb subunits; it influences the blood pH [15].

#### 2.1.3. Partial Oxygen Pressure Gradients in the Microcirculation and Interstitial Tissue

There is a pressure gradient between the PaO_2_ and PbtO_2_ (i.e., radial gradient) [16]. As mentioned in Section 2.1.1, this radial gradient of partial oxygen pressure across the microcirculation’s vessels’ walls drives the diffusion of O_2_ outside the vessels [16,17]. The values of PbtO_2_ are the net result of the O_2_ inflow into the interstitial tissue from the vessels and the O_2_ outflow from the interstitial tissue into the mitochondria for consumption by the electron transport chain [12,17].

The O_2_ outflow from the microcirculation starts from the arterioles and leads to a progressive decrease of PaO_2_ toward the capillaries [17,18,19,20]. Micro-vessels exchange O_2_ via convective flow and diffusion (when there is a differential PaO_2_ in nearby micro-vessels) [21,22]. The O_2_ diffusion includes backflow from areas of the interstitial tissue with higher PbtO_2_ than PaO_2_ [21,22]. The venoules receive poorly oxygenated blood from the capillaries together with well-oxygenated blood from the arterioles through arteriovenous shunts, resulting in a higher level of PaO_2_ in the venoules compared to that in the capillaries [19,23]. This complex O_2_ transport across the microcirculation results in a U-shaped PaO_2_ longitudinal gradient, with the lowest levels in the capillaries [16,19].

The non-uniform PaO_2_ longitudinal and radial gradients along the microcirculation imply a heterogeneity of the O_2_ diffusion into the interstitial tissue. The PbtO_2_ changes progressively between the vessels and the cells, according to both the proximity of the interstitial tissue to the vessels of the microcirculation, and to their type (e.g., arterioles, capillaries), as the O_2_ diffuses at different rates between the vessels and the mitochondria [12,17,19,20,24]. This also implies that the levels of oxygenation in the cell vary with the distance to the vessels [12].

In healthy brain tissue, the PaO_2_ longitudinal gradient is flatter than in the other tissues (e.g., resting skeletal muscle) due to the high brain metabolism, which drives oxygen passage outside vessels at all stages along the microcirculation; and the high tissue perfusion, which creates a high blood transit velocity along the microcirculation [16]. As a result of the lesser PaO_2_ longitudinal gradient (and consequently a lesser PaO_2_ radial gradient and differences in O_2_ diffusion along the microcirculation), as well as the presence of the Virchow–Robin space, the PbtO_2_ in a healthy brain can be considered approximately uniform [23].

#### 2.1.4. Roles of the Fåhraeus Effect and Glycocalyx Layer in Oxygen Diffusion in the Microcirculation

##### 2.1.4.1. Reduction in Hematocrit Along the Microcirculation

The Fåhraeus effect is the progressive decrease of the hematocrit in the smallest vessels of the microcirculation [25,26]. Therefore, the O_2_ carried per unit cross-sectional area by the capillaries decreases compared to the arterioles, which affects the longitudinal gradient along the microcirculation [16].

##### 2.1.4.2. Plasma Gap

The volume of plasma between the erythrocytes and the vessel wall (i.e., plasma gap) is an obstacle to O_2_ diffusion, as the O_2_ has to move across it to diffuse outside the vessels. This means that if there is an increase in the plasma gap, O_2_ diffusion is reduced. Since the Fåhraeus effect increases the plasma gap along the microcirculation, the O_2_ diffusion is more profoundly affected in the smallest vessels [16].

In addition to the Fåhraeus effect, the glycocalyx layer of the capillary walls can also widen the plasma gap [16].

#### 2.1.5. Cerebral Blood Flow Autoregulation

The autoregulation of cerebral blood flow (CBF) maintains constant perfusion in the microcirculation regardless of changes in cerebral perfusion pressure (CPP), provided that CPP lies within a physiological range, by vasoconstricting or vasodilating the vessels in the microcirculation [27]. By regulating the CBF, the PbtO_2_ is held constant: the variable radii of the vessels of the microcirculation prevent any change in the amount of O_2_ delivered [28].

#### 2.1.6. Vascular Tone of the Microcirculation According to the Tissue Metabolic Status

Similar to the cerebral autoregulation described in Section 2.1.5, the brain metabolic status can also influence the radii of the vessels in the microcirculation and so the O_2_ delivered. This regulation is driven through by-products of the metabolism itself (e.g., CO_2_), or molecules released by the endothelium and/or erythrocytes in response to the status of tissue saturation (e.g., nitric oxide (NO), nitrite) [22,28,29].

It should be mentioned that the complexity of the homeostatic mechanism makes it possible to maintain the tissue O_2_ metabolism constant despite different Hb concentrations in the vessels [30].

### 2.2. Tissue Respiration in Traumatic Brain Injury

The pathogenetic mechanisms that follow brain trauma can result in several abnormalities in tissue respiration, which exhibit high interpatient and intrapatient variabilities (Section 1). Systemic comorbidities (e.g., hemorrhagic shock) and the treatment received can further exacerbate these abnormalities as well as the scale of the divergence between patients.

#### 2.2.1. Reduction of Circulatory Oxygen Delivery Capacity

The circulatory O_2_ delivery capacity to the brain is the product of the CBF and oxygen-carrying capacity per unit volume [31]. A reduction in either means less O_2_ delivered to the tissue and thus a reduction in PbtO_2_ [19]. It should be highlighted that the complexity of brain trauma pathogenesis, and the different effects it has on the physiological mechanisms that maintain brain homeostasis, mean it is not always possible to infer brain ischemia from a certain status of cerebral perfusion or Hb levels alone [30,32].

##### 2.2.1.1. Reduction of Cerebral Blood Flow

Brain trauma can reduce the CBF: the impairment can be localized to specific regions; its magnitude can vary across the brain; it can be related to the systemic status (e.g., pyrexia) [33,34,35]. The decreased cerebral perfusion reduces the O_2_ transported into the tissue and so decreases the PbtO_2_ by reducing the O_2_ delivery [36,37,38,39]. The relevance of CBF to tissue oxygenation post-trauma is demonstrated by the positive effect that fluid replacement and plasma expansion have on the O_2_ delivery by re-establishing the CBF without increasing (and theoretically decreasing) the oxygen‑carrying capacity per unit volume [31]. However, the pathogenesis of tissue respiration in TBI and the treatments to prevent ischemic episodes cannot be limited to abnormalities in CBF and efforts to prevent/normalize it respectively [40].

##### 2.2.1.2. Reduction of Hematocrit

Reduced hematocrit means reduced Hb concentration and in turn a reduced oxygen‑carrying capacity per unit volume.

As expected, red blood cell transfusion in TBI patients increases the oxygen‑carrying capacity, and so the PbtO_2_ the same [41]. However, it should be mentioned that increasing the hematocrit yields also an undesirable increase in viscosity, which lowers CBF and thus hampers the positive effects that the increased oxygen-carrying capacity may have otherwise had on O_2_ delivery [42,43].

##### 2.2.1.3. Response to Therapeutic Hyperoxia

Clinical studies on a TBI population showed that increasing the PaO_2_ (i.e., hyperoxia) caused the PbtO_2_ and the brain aerobic metabolism to increase [44,45].

The limited propensity of O_2_ to diffuse into liquids and the high arterial saturation in the lungs together prohibit all but small increases of total O_2_ due to hyperoxia [46].

#### 2.2.2. Metabolic Dysfunction

Following brain trauma, there can be a metabolic dysfunction characterized by impairment in the aerobic metabolism (i.e., cytopathic hypoxia) due to mitochondrial damage. Even at non‑pathological levels of PbtO_2_ and CPP, this impairment can result in anaerobic energy production (i.e., hyperglycolysis) as a compensatory mechanism [47,48,49,50]. The inability of the mitochondria to use O_2_, and the associated hyperglycolysis, decreases the O_2_ consumption and increases the lactate/pyruvate ratio, respectively [50,51]. Metabolic dysfunction can be unrelated to the tissue perfusion [32,52,53].

The relevance of metabolic dysfunction to tissue respiration can be appreciated in the absence of a metabolic response (e.g., normalization of the lactate/pyruvate ratio) when O_2_ delivery is elevated via increased hematocrit or hyperoxia, as described in Section 2.2.1.2, Section 2.2.1.3, despite an increase in PbtO_2_ [54,55,56]. The results from the clinical studies described in Section 2.2.1.3, which reported a reduction in the lactate/pyruvate ratio as a response to hyperoxia, were attributed to the progressive deterioration of the metabolic dysfunction, which made the hyperoxia most effective in the early hours from injury, and to O_2_ extraction by a percentage of surviving mitochondria [44,45]. The different responses to hyperoxia across clinical studies highlight the interpatient variability of the brain metabolic dysfunction after trauma and the significance of the time since injury.

It should be stressed that the metabolic dysfunction is a significant component in TBI pathogenesis, and an increase in lactate/pyruvate ratio is linked to poorer clinical outcomes [57,58].

#### 2.2.3. Microcirculatory Dysfunction

There could be multiple structural impairments to the microcirculation that give rise to abnormal O_2_ diffusion and significantly alter the PbtO_2_ [59]. Analogous to abnormal O_2_ consumption, as explained in Section 2.2.2, abnormal O_2_ diffusion can also affect the tissue respiration in normally perfused brain regions [9].

##### 2.2.3.1. Anatomical Damage to the Vessels in the Microcirculation

The end organ impairment in TBI patients can be influenced by multiple systemic and local structural injury factors, which results in an impairment in the O_2_ diffusion: after brain trauma, there can be endothelium swelling, BBB damage, and perivascular edema [60]. A recent study reported that hemorrhagic shock trauma patients may have a systemic endotheliopathy in the early hours after injury, which is characterized by endothelial cell damage and glycocalyx shedding, and which leads to a reduction of blood flow and vascular density in the microcirculation (Section 2.2.3.2) [61].

##### 2.2.3.2. Reduction of Vascular Density in the Microcirculation

The distance between cells and their nearest vessels affects the rate at which O_2_ diffuses into the cells, as well as the O_2_ concentration in the cells and the interstitial tissue: O_2_ must travel this distances and progressively diffuse across various tissue sizes [11,62]. After brain trauma, some vessels in the microcirculation become occluded or collapse, increasing the distance between the center of the perfused capillaries and the farthest cells that receive their O_2_ [60]. This reduction of vascular density decreases PbtO_2_ and lowers aerobic metabolism, with the greatest impact in the tissue farthest from the perfused capillaries [60,63,64]. The relevance of vascular density to PbtO_2_ and tissue respiration is illustrated by the lack of response in the PbtO_2_ and O_2_ extraction in cases of hyperoxia when the vascular density is low [46].

An increase in CPP can reopen occluded capillaries and increase O_2_ diffusion [19,65].

#### 2.2.4. Abnormalities in Cerebrovascular Regulation

Cerebrovascular regulation (Section 2.1.5) can be impaired in some TBI patients, so that changes in CPP translate directly into changes in CBF [66].

#### 2.2.5. Cerebral Vasospasm

Cerebral perfusion in TBI patients can be impaired by episodes of vasospasm caused by a secondary injury [67].

#### 2.2.6. Abnormalities in the Microcirculatory Reactivity to the Metabolic Status

The homeostatic role of microcirculatory regulation in cell metabolism explained in Section 2.1.6 can be impaired in TBI patients. For example, the compartmentalization of NO by erythrocytes (Section 2.1.6) could be absent in patients who receive a blood transfusion with consequent vasodilation [42].

The relevance of CBF regulation in the microcirculation by molecules indicative of metabolic status is demonstrated during therapeutic hyperventilation to reduce ICP: prolonged hyperventilation can significantly reduce levels of CO2, which leads to ischemia by vasoconstriction [68].

#### 2.2.7. Fåhraeus Effect and the Role of Endotheliopathy in Brain Trauma Microcirculation

The pathophysiological mechanism of brain trauma can act via the Fåhraeus effect to change the physiology of tissue respiration. Impaired regulation of the vessels’ radii in the microcirculation due to brain trauma, and/or a reduction of hematocrit due to hemorrhage, can both theoretically induce sooner and exacerbate the physiological decrease in hematocrit along the microcirculation described in Section 2.1.4. This perturbs the physiological oxygen-carrying capacity along the microcirculation described in Section 2.1.4.1. Furthermore, an increased plasma gap reduces O2 diffusion (Section 2.1.4) and compartmentalization of NO by erythrocytes (Section 2.1.6) [29]. Damage to the endothelial glycocalyx (e.g., endotheliopathy (Section 2.2.3.1)) can also affect the plasma gap.

#### 2.2.8. Shifts in the Oxygen–Hemoglobin Dissociation Curve in the Tissue Microcirculation

The heterotropic allosteric modulators shift the oxygen–hemoglobin dissociation curve to the right to different degrees, depending on the changes in their concentrations after brain trauma. The resulting modulations of PaO_2_ in the physiological microcirculation are significant because the dissociation curve has a sigmoidal shape, and the PaO_2_ approaches the steepest region of the curve. In brain trauma, the modulators are even more relevant to the tissue respiration, because they regulate the curve at even steeper segments due to further reductions in PaO_2_.

Systemic changes related to systemic comorbidities, such as respiratory acidosis or sepsis, could further compound locally abnormal modulator concentrations [69]. Akin to the clinical status, different treatments and/or the response to treatments can alter the modulators’ concentrations and therefore contribute to the interpatient variability.

##### 2.2.8.1. Hydrogen Concentration

The severity of the impairments to tissue metabolism described in Section 2.2.2 can lower proportionally the pH in the interstitial tissue [70,71]. The H^+^ concentration in the erythrocytes, which regulates the oxygen–hemoglobin dissociation curve, is in equilibrium with that of the plasma and, in turn, that of the interstitial tissue [72]. Therefore, the lowered pH in the interstitial tissue facilitates the Hb dissociation to O_2_ in the erythrocytes. Decreased Hb concentration due to hemorrhage after trauma can further increase the H^+^ concentration via reduced Hb buffering.

Similar to the metabolic impairment in TBI, the interstitial tissue pH is associated with the patient’s prognosis, as the two phenomena are linked [73,74].

##### 2.2.8.2. Carbon Dioxide Concentration

Tissue acidosis due to metabolic impairment and the reduction of CO2 clearance owing to low perfusion increase the tissue CO2 concentration (or partial pressure) [70]. As discussed in Section 2.2.8.1, CO2 concentration is related to patients’ prognoses [70].

##### 2.2.8.3. 2,3-Diphosphoglycerate Concentration

Pathogenetic and therapeutic components can modify the concentration of 2,3-DPG in TBI patients: systemic inflammatory response syndrome and/or nutritional support in severely ill patients (e.g., malnutrition; a history of drug abuse) can lead to hypophosphatemia, which reduces levels of 2,3-DPG [69,75,76]; the transfusion of stored red blood cells can result in lower levels of 2,3-DPG than of fresh red blood cells [77].

High levels of 2,3-DPG were reported in the plasma after brain trauma [78]. However, the relationship between the plasma—and erythrocytes—2,3-DPG after brain trauma and their combined effect on cerebral tissue respiration are still under investigation [78].

##### 2.2.8.4. Chloride Concentration

Significant infusion of saline 0.9% to resuscitate shocked trauma patients can result in hyperchloremic acidosis [79,80]; this condition can shift the oxygen–hemoglobin dissociation curve by changing the blood pH and/or by allosteric modulation (with the former mechanism more profound than the latter).

The use of other drugs to resuscitate patients can avoid hyperchloremic acidosis [81,82]. This highlights the interpatient variability of tissue respiration due to the systemic status and the treatment applied.

##### 2.2.8.5. Temperature

After brain trauma, there is generally an increase in the brain temperature, which can exceed that of the core, which can be already in a pyretic status [83]. The temperature delta between the brain and the core further decreases oxygen–hemoglobin affinity (already changed by the high core temperature), when blood perfuses into the brain tissue [83,84].

Hyperthermia is associated with poor outcomes and brain damage, including to the BBB and endothelial cells, and metabolic dysfunction [84,85,86]. The extent of this damage in a conserved cerebral perfused status can be limited [87].

Therapeutic interventions to lower the temperature in TBI patients could reduce PbtO_2_ via physiological thermoregulatory responses (e.g., shivering) [88].

## 3. Intracranial Tissue Partial Oxygen Pressure and Near-Infrared Spectroscopy Neuromonitoring in Acute Traumatic Brain Injury

A PbtO_2_ of <20 mmHg and <10 mmHg characterize respectively mild brain hypoxia and severe brain hypoxia; they are associated with poor outcomes and high mortality rates [89,90]. Clinical studies reported improved outcomes and a reduced mortality rate when intracranial PbtO_2_ neuromonitoring is included in the clinical decision process to avoid brain hypoxia [89,91,92,93,94].

The link between PbtO_2_ and tissue respiration, the thresholds linked to ischemia, and the efficacy of PbtO_2_-guided therapy collectively led toward the assessment of the efficacy of NIRS to detect episodes of ischemia: absolute values of tissue saturation were compared with PbtO_2_; and trends of changes in tissue saturation were compared with PbtO_2_ oscillations (especially those within the ischemic window) [4,5]. However, a direct comparison between the values reported by the two techniques can be distorted by abnormalities in the tissue respiration related to the pathogenesis of brain trauma.

### 3.1. Inability to Infer the Whole Pathological Tissue Respiration Status by Analysis of Tissue Partial Oxygen Pressure Alone

#### 3.1.1. The Interstitial Tissue Partial Oxygen Pressure Neuromonitoring Does Not Respond to All Pathological Abnormalities in Tissue Respiration

PbtO_2_ does not represent all the pathophysiological mechanisms that can be impaired in brain trauma. In the absence of oxygen consumption by cells due to metabolic impairments, the oxygen diffuses from the vessels into the interstitial tissue, until equilibrium of partial oxygen pressure is reached between the vessels and interstitial tissue. Consequently, high/normal PbtO_2_ may hide an impairment of tissue respiration in brain cells [51]. The extent to which the PbtO_2_ and the respiration at the cellular level are uncorrelated would vary at different grades of severity of metabolic dysfunction.

The importance of metabolic impairment to the assessment of tissue respiration can be appreciated when the ischemic thresholds of PbtO_2_ for stroke patients are applied to TBI patients. These thresholds are not transferable, as the latter patients present a metabolic impairment that is not present in the former [95].

#### 3.1.2. The Interstitial Tissue Partial Oxygen Pressure Is an Average of Different Pathogenetic Mechanisms Related to Brain Trauma

PbtO_2_ is the product of multiple processes of tissue respiration, each of which can be impaired by multiple pathogenetic variables (e.g., cerebral perfusion, vascular density, BBB integrity). Therefore, the effects of each pathogenetic mechanism on each process in the tissue respiration cannot be inferred from the PbtO_2_ alone.

Some pathological mechanisms of brain trauma can increase PbtO_2_ (e.g., lower O_2_ affinity in the oxygen–hemoglobin dissociation curve), while others can reduce it (e.g., increase in plasma gap, microcirculation damage, reduction of hematocrit, reduction in vascular density). PbtO_2_ is a weighted average of the effects of multiple pathogenetic mechanisms, each of which affects its value with different magnitude and sign. Moreover, there is significant variability in the relevance of each pathogenetic mechanism to tissue respiration across patients, and for the same patient at different times since injury.

This could explain the conflicting attempts in the literature to attribute abnormal values of PbtO_2_ primarily to a specific pathogenetic mechanism, as well as the lack of correlation between PbtO_2_ and other pathophysiological parameters (e.g., CBF, ICP, lactate/pyruvate ratio) reported by some studies [59,63,96,97,98].

### 3.2. The Intracranial Tissue Partial Oxygen Pressure and Near-Infrared Spectroscopy Neuromonitoring Can Be Affected Differently by the Brain Trauma Pathogenesis

The PbtO_2_ and NIRS neuromonitoring analyze two separate components of the tissue respiration in two separate volumes: gaseous O_2_ in the interstitial tissue, and oxyhemoglobin (O_2_Hb) and deoxyhemoglobin (HHb) concentrations in the vessels, respectively. Since the objects of investigation by the two techniques can be affected differently by the brain trauma pathogenesis, so can the values reported by PbtO_2_ and NIRS neuromonitoring. This disparity can result in different trends, degrees of change, or times of response between the two values [99,100]. The distinct projections of tissue respiration by PbtO_2_ and NIRS neuromonitoring can be further enhanced by the different samples of tissue analyzed by each technique and by the heterogeneity of the pathogenetic mechanism of trauma across the brain.

In summary, the mixed roles of multiple pathogenetic mechanisms to determine the tissue respiration may produce disagreement between the values reported by PbtO_2_ and NIRS neuromonitoring.

#### 3.2.1. Metabolic Dysfunction

The high PbtO_2_ during the metabolic dysfunction described in Section 3.1.1 can be coupled with high tissue saturation. Due to the low O_2_ diffusion into the interstitial tissue, only a small proportion of the total O_2_ carried is required such that PaO_2_ reaches equilibrium with PbtO_2_. Thus, if the O_2_ consumption by tissues is impaired by trauma, the amount of O_2_ that must be unbound from the Hb to reach this equilibrium is extremely low, resulting in a high ratio of O_2_Hb/total Hb.

As mentioned in Section 3.1.1, different degrees of metabolic dysfunction can disturb the PbtO_2_ and NIRS values to a greater or lesser extent. A partial metabolic dysfunction would maintain a constant low flow of O_2_ from the vessels into the cells, so that O_2_ is unbound from Hb at a lesser rate. This would not be the case if there were a complete metabolic dysfunction in the cells.

As mentioned in Section 3.2, since the pathological tissue respiration comprises multiple components other than the metabolic dysfunction, the effect that this may have on PbtO_2_ and NIRS values varies.

#### 3.2.2. Reduction in Oxygen Diffusion and Oxygen‑Carrying Capacity

The values reported through PbtO_2_ and NIRS neuromonitoring could be affected, to a greater or lesser extent, by the different relevance of the pathological O_2_ diffusion and oxygen‑carrying capacity to the brain status.

Impairments to O_2_ diffusion (e.g., anatomical damage to the microcirculation) without a corresponding reduction in tissue perfusion can lower PbtO_2_ without a proportionate reduction in tissue saturation, and vice versa. Furthermore, in those areas with intact vasogenic response, the increased CO_2_ concentration (Section 2.2.8.2) and the different diffusion capacities of O_2_ and CO_2_ would make the latter capable to vasodilate even in the context of impaired O_2_ diffusion [12]. This further increases the gap between tissue saturation and PbtO_2_.

Reductions in perfusion, hematocrit, or vascular density can decrease the number of illuminated chromophores. Thus, small changes in their concentration would appear more significant than their real, absolute changes by relative NIRS neuromonitoring. This creates a disparity between the absolute changes captured by PbtO_2_ neuromonitoring, and the relative changes seen by NIRS neuromonitoring. The relevance of a reduction in perfusion and/or vascular density to the interpretation of the relative change of optical signal can also be applied to the monitoring of changes of intravascular exogenous chromophores (e.g., indocyanine green (ICG)) in contrast-enhanced NIRS.

#### 3.2.3. Right-Shift of the Oxygen–Hemoglobin Dissociation Curve

The grades of right-shift of the oxygen–hemoglobin dissociation curve change the relationship between arterial saturation and PaO_2_: thus, the concentrations of the modulators influence the relationship between NIRS and PbtO_2_ neuromonitoring.

Similar to the comparison between NIRS and PbtO_2_ neuromonitoring, the comparative analysis of tissue saturation between patients and/or brain areas is influenced by the reduction in O_2_Hb/total Hb ratio due to the decrease in oxygen affinity [101].

### 3.3. Different Volumes and Statuses Are Analyzed by the Tissue Partial Oxygen Pressure and Near-Infrared Spectroscopy Neuromonitoring

The brain regions scanned by the intracranial PbtO_2_ monitor and the NIRS probes applied on the scalp are different: the intracranial PbtO_2_ catheter observes an area predominantly in the white matter, whereas the NIRS probes scan at 3 cm depth from the scalp, comprising the extracranial tissue, the skull, the arterial vessels, the venous sinuses, the cerebral spinal fluid, and the most superficial part of the brain (predominantly neocortex) [20,102,103]. The different volumes scanned lead to different reported values due to the pathogenesis of the trauma and/or therapeutic interventions.

#### 3.3.1. Different Tissue Statuses Across the Brain Can Influence the Values Reported

Clinical studies reported different statuses across the brain due to brain damage (e.g., CPP, PbtO_2_, lactate/pyruvate ratio) [33,104,105]. Due to this anatomical mosaic of statuses across the brain and the different volumes investigated by PbtO_2_ and NIRS neuromonitoring, their associated values are indicative of different brain statuses (as well as depend on separate confounding factors). Changes of the values reported by each technique can be ascribed, partially or completely, to different statuses, evolution of the pathology (e.g., increase of brain edema), and response to therapeutic interventions, such as decompressive craniectomy, between the areas scanned (Figure 1 and Figure 2).

#### 3.3.2. Changes in the Volumes Examined

Anatomical lesions, progression of the disease, and therapeutic interventions (e.g., decompressive craniectomy) can all modify the regions scanned by the two techniques (Figure 1 and Figure 2). Changes of the tissue optical properties alter the optical path length, extracranial and intracranial hemorrhages, significantly increase light absorption (Figure 1B,C) [106]; fluctuations in O_2_Hb and HHb concentrations in the microcirculation related to pathogenetic mechanisms (e.g., reduction of cerebral blood flow, reduction of vascular density, abnormal cerebrovascular regulation, shifts to the oxygen–hemoglobin dissociation curve) do the same; therapeutic interventions perturb the scattering and absorption coefficients by changing the illuminated layers (e.g., increase of volume of the venous sinuses, expansion of brain edema) and removing and/or adding new layers along the optical path length (e.g., removal of the skull, addition of air) (Figure 2B,C) [107,108,109]. The areas scanned by PbtO_2_ neuromonitoring can also vary between recordings taken before and after therapeutic interventions; for example, when the brain shifts due to reduction of midline shift after a decompressive craniectomy (Figure 2A).

The fact that the values reported are affected by any changes to the volumes examined compounds the discrepancy between the outputs of PbtO_2_ and NIRS neuromonitoring, both between patients and for the same patient at different times since injury; furthermore, the comparison of recordings made by the same technique across patients and times since injury is affected.

It should be mentioned that the significant light absorption caused by hemorrhages can be used to diagnose this type of injury [110,111].

### 3.4. Heterogeneity of Values within the Volume Analyzed by the Tissue Partial Oxygen Pressure and Near‑Infrared Spectroscopy Neuromonitoring

The values reported by PbtO_2_ and NIRS neuromonitoring are an average within the respective volumes analyzed. The breadth of the true spectrum of values can increase due to the pathogenesis of brain trauma; consequently, misleading conclusions can be drawn on the status of the tissue analyzed (partial volume effect).

#### 3.4.1. Tissue Partial Oxygen Pressure Neuromonitoring

The reduction in aerobic metabolism and perfusion can increase the longitudinal gradient along the microcirculation and produce non-uniform PbtO_2_. Its value is a weighted average of the partial oxygen pressure in an area of the interstitial tissue perfused by hundreds of micro-vessels [19,103]. The poor oxygenation of cells in the vicinity of the lowest PbtO_2_ would be averaged with the better oxygenation of cells surrounded by higher PbtO_2_ [24].

The variability of the properties of the sample area is also increased by anatomical damage to the microcirculation (e.g., necrotic areas, vasogenic and cytotoxic edema) or systemic abnormalities (e.g., hemorrhagic shock), which increase the non-uniformity of PbtO_2_ by impairing perfusion [112].

#### 3.4.2. Near-Infrared Spectroscopy Neuromonitoring

The illuminated tissue comprises multiple micro-vessels. Different levels of oxygenation across the microcirculation are averaged into a single value of tissue saturation. Akin to Section 3.4.1, the extent to which some micro-vessels are poorly oxygenated can be ignored or underestimated because their signal is averaged with that of more oxygenated vessels.

Furthermore, the illuminated volume comprises vessels from both the arterial and venous systems. Anatomical damage to the microcirculation and a reduction of vascular density alter the oxygenation, as well as the relative number, of illuminated arterial and venous vessels.

### 3.5. Barriers to Accurate Data Acquisition Using Tissue Partial Oxygen Pressure and Near-Infrared Spectroscopy Neuromonitoring

#### 3.5.1. Tissue Partial Oxygen Pressure Neuromonitoring

The data acquisition can be impaired in multiple ways: during the insertion of the catheter, microhemorrhages in the parenchyma can occur, which affect the reading [113]; some authors reported different PbtO_2_ based on the type of catheter, although others reported that this difference was not significant [19,104]; measurements taken during the first hour from insertion are inaccurate [90]; the vicinity of the intracranial PbtO_2_ monitor may become infected, voiding the measurements and demanding that the catheter be removed, and the recordings ceased.

There is no definitive agreement as to whether the PbtO_2_ monitor should be positioned to scan the healthy tissue or brain edema nor whether the catheter should be placed to monitor the gray or white matter [19,69]. The position of the catheter according to these choices can significantly change the values measured [20,114,115,116,117,118]. Clinical studies showed a correlation between the areas with metabolic dysfunction and their images on the CT scan [52]. However, there could be areas with metabolic impairments that cannot be always identified on a CT scan (e.g., perilesional tissue) (Figure 1A) [47,119]. Thus, the values recorded by the PbtO_2_ monitor could indicate healthy tissue, while in reality, the tissue scanned was affected by metabolic dysfunction to some extent. The lack of standardization regarding use of the PbtO_2_ monitor, and the inability to reliably infer the brain status from the CT scan, make for cautious interpretation of PbtO_2_ values.

#### 3.5.2. Near-Infrared Spectroscopy Neuromonitoring

The application of NIRS in a clinical context can be hampered by multiple elements that affect the quality of the data recorded: sweat, which affects the probe–skin coupling; motion artifacts, due to involuntary movements under light sedation; ambient light–noise; poor feedback on the quality of the signal acquired during the acquisition; etc. [7,120,121].

The extracranial tissue (ECT) is included in the optical path; changes in the scalp hemodynamic can influence the signal detected and impair analysis of the brain. The use of multiple channels at different source-detector (SD) distances in spatially resolved spectroscopy makes the detection and isolation of the brain hemodynamic possible, overcoming the confounding factor of the ECT (Figure 1B) [122,123].

The differential pathlength factor (DPF) used in physiological tissue is sometimes not applicable in TBI patients, since there are significant changes in scattering caused by the presence of air, after surgical intervention, and/or the presence of water, due to brain edema [124]. Thus, application of the modified Beer–Lambert law, to measure the concentrations of O_2_Hb and HHb, could be inaccurate in TBI patients [124]. Furthermore, changes in scattering during the data acquisition due to the evolution of the disease (e.g., expansion of the edema) can be misinterpreted as changes in the concentrations of O_2_Hb and HHb. Finally, significant changes in scattering within the illuminated volume can hamper light diffusion modelling using Monte Carlo or finite-element methods (FEM) within a tomographic reconstruction of the optical signal (Section 4.2.1.3).

## 4. Future Application of Tissue Partial Oxygen Pressure and Near-Infrared Spectroscopy Neuromonitoring in Clinical Practice

The complexity of the pathogenesis of brain trauma requires that NIRS be incorporated into a multimodal neuromonitoring; the optical values can be synthesized into biosignatures linked to different clinical presentations and clinical decisions.

### 4.1. Biosignatures

Clinical studies on the effects of therapeutic interventions on TBI patients (e.g., hypothermia, early decompressive craniectomy) showed poor results when applied indiscriminately to the whole TBI population [125,126]. One of the current, unaddressed clinical challenges in the treatment of the acute TBI population is the sub-classification of patients, with the view to identify the most suitable treatment for that specific group [127,128]. A deeper analysis of patient status could facilitate such classification [49,128].

Biomarkers from PbtO_2_ and NIRS neuromonitoring can be combined, together with other neuromonitoring techniques (e.g., CT; microdialysis), to scan the spectrum of abnormalities related to the brain trauma pathogenesis on tissue respiration. Multiple types of NIRS, which produce separate biomarkers, can be merged to increase this capacity. Disjointed analyses of multiple biomarkers can slow the clinical assessment and overwhelm clinicians with too many (potentially unlinked) parameters [94]. Instead, biomarkers can be used to formulate biosignatures indicative of the severity of the injury and the patient’s prognosis [129]. Combining multiple biomarkers into a single biosignature can enhance multimodal neuromonitoring: biomarkers that represent different aspects of pathogenetic mechanisms are combined for a holistic (and so more accurate) understanding of the brain status; a single parameter is used easily by clinicians to make a quick assessment and to monitor a patient’s response to treatment; a biosignature can be linked easily to patient outcomes and mortality rates and included in diagnostic/prognostic algorithms (e.g., machine learning) [129,130,131,132,133]. In summary, the use of biosignatures could help clinicians classify TBI patients based on their statuses and prognoses and then tailor their treatments accordingly.

The drugs administered (e.g., Propofol, barbiturates, 0.9% saline), comorbidities (e.g., heart failure), and the type of injury (e.g., penetrated or non-penetrated brain trauma) can each affect the parameters read (e.g., CBF, PbtO_2_) and the patient’s prognosis, to a greater or lesser extent [134,135,136]. Furthermore, the pathogenesis that impairs the tissue respiration can also affect other physiological mechanisms, such as reduced inflows of metabolites and clearance of catabolites due to reduced CBF. Biomarkers related to assessment of the tissue respiration can be merged with other elements that compound the clinical presentation in trauma patients (e.g., immune and inflammatory response) to further enhance assessment of the severity of the injury and the patient’s prognosis [137]. Thus, the parameters that comprise a biosignature should be comprehensive of other values over and above those monitored for the tissue respiration assessment: epidemiology, type of trauma, clinical presentation, systemic status, type of treatment, etc. [138].

### 4.2. Multimodal Monitoring

Herein, there is a description of a few techniques that can be merged into a multimodal neuromonitoring of TBI patients to scan for abnormalities in tissue respiration (Table 1).

It should be noted that some of these techniques allow clinicians to continuously monitor the brain status (e.g., PbtO_2_, NIRS), while others provide only a snapshot (e.g., CT) [139]. The combined analysis of these two types of technique can increase its overall accuracy (e.g., duration of an ischemic status) as well as reduce the time to respond [49,140].

The application of some of these techniques is limited to specific tertiary referral hospitals and requires training of specialized health care providers; thus, the proposed multimodal monitoring may not be the optimal solution in all clinical contexts.

#### 4.2.1. Tissue Partial Oxygen Pressure and Near-Infrared Spectroscopy Neuromonitoring

The combined assessments of the tissue respiration via PbtO_2_ and NIRS neuromonitoring can produce a clearer picture of the tissue status after brain trauma than the application of each technique separately. NIRS should not be seen as a substitute for PbtO_2_ but rather as an addition to it to enhance analysis of the tissue respiration status. The low sensitivity of NIRS to changes in brain hemodynamic is partially overcome by combining multiple types of NIRS and by formulating biosignatures out of multiple monitoring techniques (included PbtO_2_ neuromonitoring), which jointly enhance the optical signal and allow one to discard eventual inaccuracies from NIRS neuromonitoring through comparison of the optical signal with linked output parameters from other monitoring techniques (e.g., hemorrhage on CT scan) and/or NIRS techniques (e.g., perfusion status on contrast-enhanced NIRS).

To facilitate the introduction of NIRS into clinical practice, future optical research studies should be framed in specific clinical scenarios [141]: this would help clinicians to interpret the physiological meaning of the optical values and to analyze them alongside other biomarkers from separate neuromonitoring techniques; the assessment of response to treatment can be facilitated by a targeted approach to specific subgroups of TBI patients.

##### 4.2.1.1. Different Types of Near-Infrared Spectroscopy

Multiple optical techniques can be merged together to investigate different aspects of the pathophysiological process of tissue respiration related to brain injury.

Absolute and Relative Values of Tissue Saturation

To correctly assess the tissue saturation, the scattering and absorption properties of each layer of the tissue examined must be known. These can be measured directly using time-resolved NIRS (e.g., time- or frequency‑domain) or by employing spatially resolved continuous-wave NIRS fed with the scattering values from previous analysis on samples of the same type of tissue.

Rather than measure absolute values, changes in tissue saturation (i.e., changes in O_2_Hb and HHb concentrations) can be determined through the evolution over time of the absorption properties, if a scattering constant is assumed. Changes in tissue saturation can be linked to clinical challenges (e.g., hyperventilation, CO_2_ reactivity, Valsalva maneuver) to detect deviations from the physiological response, akin to other neuromonitoring techniques (e.g., PbtO_2_), or to the administration of drugs (e.g., adrenalin) [59,142,143]. By focusing on the changes in tissue saturation, any differences due to brain trauma of both the volume of the illuminated tissue and the optical properties within the sample can be avoided or mitigated.

The ECT confounding factor present when assessing the brain through absolute values of tissue saturation can be mitigated against, to a greater or lesser extent, by focusing on the changes in optical signal during clinical challenges. This is because the ECT hemodynamic can be considered constant, and therefore, any changes can be attributed mainly to the brain hemodynamic.

##### 4.2.1.2. Contrast-Enhanced Near-Infrared Spectroscopy

The passage of a bolus of contrast dye (i.e., ICG) through the brain can be monitored using NIRS [120,144]. The kinetic of the ICG can be related to cerebral perfusion and BBB integrity [120,144].

The short half-life of ICG allows for repeated monitoring of the bolus’ kinetic as well as alternating assessment of dye passage with that of tissue saturation. The addition of contrast‑enhanced NIRS to the wider NIRS neuromonitoring allows clinicians to obtain a clearer picture of the tissue status by combining measurement of saturation with other pathophysiological parameters that describe tissue respiration.

The use of contrast‑enhanced NIRS would not necessarily require the measurement of absolute optical properties in the illuminated tissue and/or absolute values of physiological parameters (e.g., CBF) to assess the brain status [144].

##### 4.2.1.3. Diffuse Optical Tomography

Diffuse optical tomography (DOT) uses a distributed high-density array of NIRS probes (overlapping channels with different SD distances), models of light propagation within tissue, and inverse approaches to analyze the illuminated volumes and thus obtain the three‑dimensional spatial distribution of optical properties on a co-registered structural image (e.g., CT, magnetic resonance imaging [MRI]) (Figure 1B–E) [102,145,146].

The use of an array of overlapping channels at different SD distances enhances the image resolution of the optical signal [147]. The inaccuracies caused by the partial volume effect can also be addressed through the spatial resolution within the illuminated volume [147]. Furthermore, DOT‑NIRS offers the advantage of addressing the heterogeneity of the tissue status across the brain by correlating optical values from a scanned area to anatomical injuries (e.g., edema, epidural hemorrhage) (Figure 1B,C).

Similar to that explained in Section 4.2.1.2, relative analysis of the tissue saturation in DOT can be combined with contrast‑enhanced NIRS.

#### 4.2.2. Computerized Tomography and Magnetic Resonance Imaging

The radiological assessment of the CT scan is a crucial moment in TBI assessment, and it is performed on all moderate and severe TBI patients [69]. Optical research studies found a correlation between the structural image and the probes’ positions or based the latter upon the position of the lesions seen on the CT scan [106,148]. The position of the tip of the PbtO_2_ catheter in the brain must be pinpointed on a CT scan in order to link the values reported with the status of the tissue seen on the structural image [139]. In much the same way, the NIRS values should be co-registered on patients’ CT scans: to compare the values reported by the two techniques; to avoid inaccuracies and/or misinterpretations due to the heterogeneity of tissue status across the brain; and to guide the probes’ positioning to the regions of interest [145]. It is common to scan a TBI patient multiple times to track the evolution of their injury and/or their response to treatment; then, the values reported by PbtO_2_ and NIRS neuromonitoring can be updated according to the progression of either. The different, somewhat orthogonal, natures of the radiological assessment, which scans the structure of an injury, and PbtO_2_ and NIRS neuromonitoring, which scan the functional status of an injury, enhance significantly the assessment of patient status when their respective values are jointly co-registered.

Knowledge of the anatomy of the illuminated layers facilitates adaptation of the DPF for each layer accordingly. This avoids inaccuracies in the measurement of O_2_Hb and HHb concentrations due to significant changes in scattering (Section 3.5.2).

The co-registration of the optical signal with a structural image enhances the quality of the optical image [147]. This is even more relevant to the optical analysis on TBI patients, if impairments are to be limited to the Monte Carlo and FEM reconstructions (Section 3.5.2). The structural image must be analyzed in order to reconstruct the optical properties of the illuminated layers and plot the illuminated volume.

After the initial trauma evaluation and resuscitation have been completed, structural injuries (e.g., infarction, ischemic penumbra, edema, diffuse axonal injury) on TBI patients can be evaluated using MRI [69]. Similar to what was discussed about the CT scans, the optical signal can be co-registered on patients’ MRI scans.

#### 4.2.3. Microdialysis

The lactate/pyruvate ratio is a strong indicator of aerobic or anaerobic metabolism [47,149]. Thus, by integrating microdialysis into the multimodal monitoring, the metabolic status of the tissue can be better understood [150]. Similar to PbtO_2_ neuromonitoring, the values obtained through microdialysis must be associated with the position of the catheter on the CT scan, and they vary from the other pathophysiological parameters to some degree [45,139].

#### 4.2.4. Mean Arterial Pressure and Intracranial Pressure Monitoring

The measurements of mean arterial pressure (MAP), ICP, and CPP (difference between MAP and ICP) result in an estimation of the amount of oxygen delivered and the status of cerebral autoregulation [66,151]. The complexity of the clinical status in TBI and the heterogeneity of ICP across the brain after trauma prevent clinicians from assessing the tissue respiration by analyses of these parameters alone [33,98].

#### 4.2.5. Arterial Blood Gas Analysis

TBI patients can exhibit systemic responses to trauma (e.g., increase in blood lactate) and have comorbidities that alter the physiological blood pH (e.g., acute respiratory distress syndrome (ARDS)) and electrolyte balance, such as hyper- and hyponatremia due to diabetes insipidus and syndrome of inappropriate antidiuretic hormone secretion (SIADH), respectively [152,153,154]. These systemic abnormalities can be co-factors in the shift of the oxygen–hemoglobin dissociation curve at the tissue level: systemic-blood pH is combined with local-blood pH; the presence of acidosis due to hyperchloremia depends on the level of natremia [79]. The ABG is the only means to measure arterial pH; thus, it is a significant component in the analysis of PbtO_2_ and NIRS values [155]. The effect of the increase in blood lactate on acidosis can be monitored by calculating the anion gap [155].

#### 4.2.6. Blood Sampling

The hematocrit and electrolyte imbalance are assessed easily through blood tests and give important information regarding the correct interpretation of the values reported by PbtO_2_ and NIRS neuromonitoring (Section 2.2.1.2, Section 4.2.5).

Via the Donnan effect, the higher intracellular concentration of non-diffusible anions (e.g., 2,3‑DPG) lowers the pH in the erythrocyte cytoplasm by ≈0.2 pH units compared to the plasma [156]. An increase in plasma‑2,3‑DPG, as reported in Section 2.2.8.3, uncompensated by an equal increase of non‑diffusible anions in the erythrocytes, can mitigate the Donnan effect. On the other hand, a reduction in plasma proteins can exacerbate the Donnan effect [157,158]. Thus, comparative analysis of PbtO_2_ and tissue saturation at a given blood pH may theoretically carry a small degree of error. It should be noted that the Donnan effect is mainly due to the nucleic acids and proteins concentrations in the cytoplasm, and so a decrease in erythrocytes‑2,3‑DPG, as described in Section 2.2.8.3, would not significantly change the Donnan effect.

## 5. Conclusions

The complexity and nature of abnormalities in tissue respiration due to brain trauma make PbtO_2_ insufficient to encapsulate alone and in full all aspects of the evolving pathophysiology. Thus, attempts to determine the capabilities and applicability of NIRS as a valid neuromonitoring tool based solely on comparison with PbtO_2_ neuromonitoring are conceivably sub-optimal. The pathogenesis of brain trauma suggests instead that a different approach, based on the development of biosignatures from a multimodal monitoring that includes NIRS, should be employed.

## Figures and Tables

**Figure 1 ijms-22-01122-f001:**
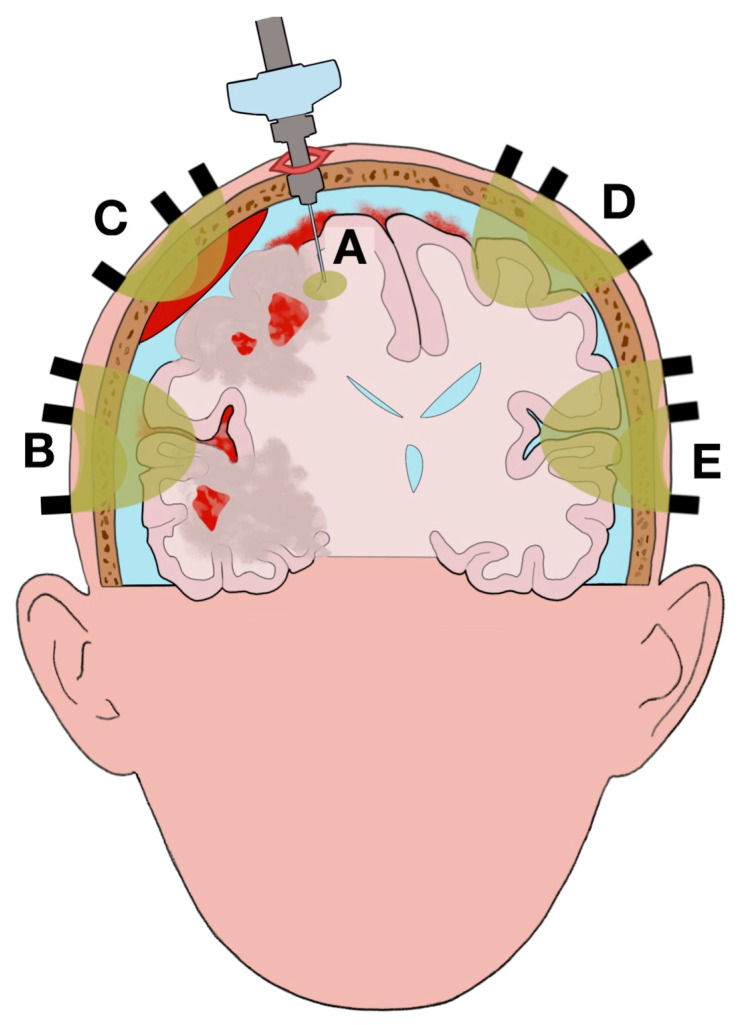
Illustration of the complexity of the pathophysiological elements related to brain trauma on the different areas scanned (highlighted in yellow) by partial oxygen pressure (PbtO_2_) and near-infrared spectroscopy (NIRS) neuromonitoring. The figure illustrates a possible clinical scenario in traumatic brain injury (TBI): epidural hematoma, subarachnoid hemorrhage, brain edema, and multiple contusions with perilesional tissue (penumbra of brain edema) on the right hemisphere causing midline shift and compression of the right lateral and third ventricles. (**A**): area scanned by intracranial PbtO_2_ monitor. The area scanned is mainly without anatomical lesions; however, a small portion is included in perilesional tissue. The heterogeneity of the brain status across the scanned area can influence the values recorded and mislead clinicians. (**B**): NIRS recording (with overlapping channels at different source-detector (SD) distances) of brain edema and subarachnoid hemorrhage. The lesions are not fully scanned because of the limited depth of the optical assessment. (**C**): NIRS recording of epidural hemorrhage. Significant light absorption by the epidural hemorrhage reduces the depth of the optical assessment and makes it impossible to monitor the brain edema and the contusions underneath. (**D**), (**E**): NIRS recording of spared tissue (normal tissue respiration). The comparison between channels B, C, D and E allows clinicians to identify the side of the lesions between hemispheres.

**Figure 2 ijms-22-01122-f002:**
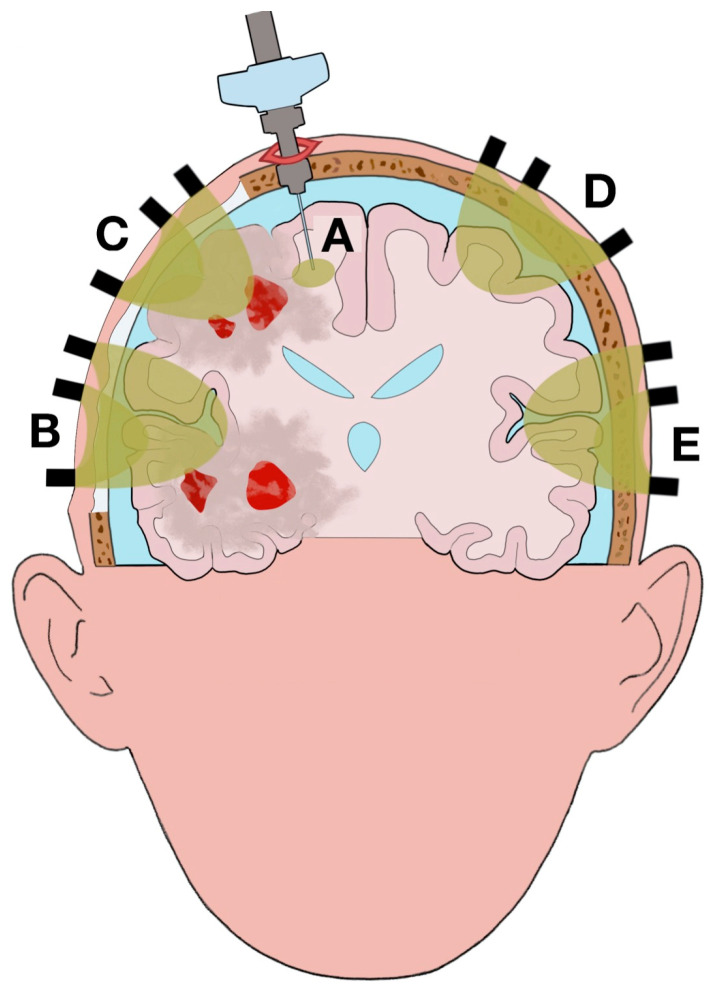
Illustration of the changes in brain statuses and scanned volumes using the two techniques after clinical evolution of the case illustrated in Figure 1. Portion of the skull is removed by the right decompressive craniectomy; air is present between the extracranial tissue (ECT) and the dura mater due to the surgical operation. Although the brain is swollen, the midline shift and the compression on the right lateral and third ventricles are reduced with a normalization of the intracranial pressure (ICP). The size of the brain edema has increased, and a new contusion has appeared due to the progression of the injury. (**A**): Area scanned by intracranial PbtO_2_ monitor. Although the position of the intracranial PbtO_2_ catheter has not changed in relation to the cranial landmarks, the scanned area is different from the one illustrated in Figure 1 due to the brain shift after decompressive craniectomy. (**B**): NIRS recording of brain edema. The absence of hemorrhage and skull, the addition of air, and the increase of the edema increased the volume recorded. (**C**): NIRS recording of brain edema and contusion. Similar to channel B, there is an increase in the volume recorded. However, the contusions present in the illuminated tissue result in a smaller volume of channel C compared to B. (**D**), (**E**): NIRS recording of spared tissue (normal tissue respiration).

**Table 1 ijms-22-01122-t001:** Techniques for multimodal monitoring of brain tissue respiration in acute and subacute, moderate, and severe TBI patients.

Type	Time	Region Monitored	Parameters
PbtO_2_ ^1^ monitor	Continuous	Brain	PbtO_2_
NIRS ^2^	Continuous	Brain	O_2_Hb ^3^, HHb ^4^
Contrast-enhanced NIRS	Intermittent	Brain	ICG ^5^
CT ^6^ head	Intermittent	Brain	Structural injuries
MRI ^7^ head	Intermittent	Brain	Structural injuries
Microdialysis	Intermittent	Brain	Lactate/pyruvate ratio
Arterial cannulation	Continuous	Systemic	MAP ^8^
ICP ^9^ monitor	Continuous	Brain	ICP
ABG ^10^	Intermittent	Systemic	Arterial pH, anion gap
Blood sample	Intermittent	Systemic	Hematocrit, electrolytes, plasma proteins, 2,3-DPG ^11^

^1^ tissue partial oxygen pressure, ^2^ near-infrared spectroscopy, ^3^ indocyanine green, ^4^ oxyhemoglobin, ^5^ deoxyhemoglobin, ^6^ computerized tomography, ^7^ magnetic resonance imaging, ^8^ mean arterial pressure, ^9^ intracranial pressure, ^10^ arterial blood gas analysis, ^11^ 2,3-diphosphoglycerate.

## Data Availability

Not applicable.

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
