# Peer review of "Mismatch between Tissue Partial Oxygen Pressure and Near-Infrared Spectroscopy Neuromonitoring of Tissue Respiration in Acute Brain Trauma: The Rationale for Implementing a Multimodal Monitoring Strategy"

_ijms, 2021, doi:10.3390/ijms22031122_

Round 1

Reviewer 1 Report

The following suggestions should be considered by the authors:

1. In the introduction the extra  hyphens should be removed.

2. What is Sec 0 , which referred many times in the text? Is this part missing from the review?

3. On page 4 §2.1.4. the notion ‘Reduction of plasma/hematocrit ratio along the microcirculation’ makes no sense.

4. The paper contains quite a lot of formatting errors (page 11, two times in legend to Fig.2, two times page 14, page 15, two times page 18, ) -  ‘Error! Reference source not found.’ – which should be removed.

5. Some paragraphs consist of one sentence only: e.g. 2.2.9. Abnormalities to cerebrovascular regulation, 2.2.10. Cerebral vasospasm, 4.2.3. Different types of near-infrared spectroscopy. They should be combined with other parts.

6. The correlation of PbtO2 and NIRS data to MRI and MR spectroscopy data broadly used in the recent clinical practice has not been discussed, CT is for brain imaging a rather outdated approach.

Author Response

Please see the PDF attached.

Reviewer 2 Report

The paper discusses an important subject. The advances in the data-acquiring process and the large extend of systems necessitates having efficient tools, such as the one proposed in this draft. The paper is well written and organized. My comments are as follows:

- While the abstract has aimed to provide a comprehensive overview of the main contribution, there is a need to be revised so that the general reader can grasp the main idea/topic of the draft and the main contribution. 

- Although a good discussion about the proposed framework's superiority is provided in terms of the numerical results, discussion about the proposed framework's complexity and how it compares with the existing techniques are highly recommended. 

- Having a nice schematic diagram in the draft would be helpful. This alleviates the difficulty of going to details of the techniques for the readers.

- There has been a surge in the application of Machine Learning and Statistical framework to solve similar problems focused in this paper. The authors are encouraged to include some of the recent articles in the introduction to give an excellent holistic overview of the existing techniques to general readers:

# "Secondary brain injury: predicting and preventing insults." Neuropharmacology 145 (2019): 145-152.

# "Optimal Finite-Horizon Perturbation Policy for Inference of Gene Regulatory Networks." IEEE Intelligent Systems (2020).

# "Machine learning multivariate pattern analysis predicts classification of posttraumatic stress disorder and its dissociative subtype: a multimodal neuroimaging approach." Psychological medicine 49.12 (2019): 2049-2059.

- The format of some of the references is not in standard form. These need to be checked and fixed.

Author Response

Please see the PDF attached.

Round 2

Reviewer 1 Report

Some of my comments have been addressed, but regarding point 6 I cannot accept the notion of the authors that "Currently the MRI is not routinely recommended as neuromonitoring tool in the acute phases of moderate and severe TBI." In the neurological clinical practice of a university hospital (at least in Western Europe) for detection of brain ischemia MRI (DWI or flair sequences) is usually performed. At least this point should be mentioned in a short paragraph, otherwise the review is simply not complete.

Author Response

Thanks for your comment and for reviewing our paper. We added the MRI as a neuromonitoring tool and we edited paragraph 4.2.2. and Table 1 accordingly.

Reviewer 2 Report

The paper is well-revised, and in my opinion, it is ready for publication. 

Author Response

Thanks for reviewing our paper.